# Transcriptomic analysis of biofilm formation in strains of *Clostridioides difficile* associated with recurrent and non-recurrent infection reveals potential candidate markers for recurrence

**Daira Rubio-Mendoza**[1], **Carlos Córdova-Fletes**[1], **Adrián Martínez-Meléndez**[2], **Rayo Morfín-Otero**[3], **Héctor Jesús Maldonado-Garza**[4], **Elvira Garza-González**[1]*

1 Departamento de Bioquímica y Medicina Molecular, Universidad Autónoma de Nuevo León, Facultad de Medicina, Monterrey, N.L., Mexico, 2 Facultad de Ciencias Químicas, Universidad Autónoma de Nuevo León, San Nicolás de los Garza, Nuevo León, Mexico, 3 Instituto de Patología Infecciosa y Experimental "Dr. Francisco Ruiz Sánchez", Centro Universitario Ciencias de la Salud, Universidad de Guadalajara, Guadalajara, Jalisco, Mexico, 4 Servicio de Gastroenterología, Hospital Universitario Dr. José Eleuterio González, Universidad Autónoma de Nuevo León, Facultad de Medicina, Monterrey, N.L., Mexico

* elvira_garza_gzz@yahoo.com

## Abstract

The transcriptomic profile in a biofilm model of ribotypes (RT) 001 and 027 associated with recurrent *Clostridioides difficile* infection (R-CDI) and not associated with recurrent (NR)-CDI was analyzed to identify genes that may favor the recurrence. Twenty strains were selected, 10 RT001 and 10 RT027. From each ribotype, 5 were R-CDI and 5 NR-CDI. Biofilm and nonadherent cells were prepared from each clinical isolate, and the RNA was extracted. RNA samples were pooled in 8 combinations implying ribotype, recurrence, and biofilm formation. Each pool was separately labeled with Cy3 dye and hybridized on a microarray designed for this study. Slides were scanned, analyzed, and gene expression was compared between unique and grouped pools using the Student's t-test with Benjamini-Hochberg correction when appropriate. Validation was carried out by qRT-PCR for selected genes. Results: After comparisons of differentially expressed genes from both ribotypes of R-CDI strains (nonadherent cells vs. biofilm) and both ribotypes in biofilm (R-CDI vs. NR-CDI), we found 3 genes over-expressed and 1 under-expressed in common (adj. $p \leq 0.05$). Overexpressed genes were *CAJ70148* (a putative dehydrogenase), *CAJ68100* (a secretion type II system protein from the GspH (pseudopilins) family), and *CAJ69725* (a putative membrane protein); under-expressed was *CAJ68151* (a segregation and condensation protein A). Because *CAJ70148*, *CAJ68100*, *CAJ69725* and *CAJ68151* were differentially expressed in biofilm in strains associated with R-CDI, they may support the biofilm favoring the recurrence of CDI. However, further studies will be needed for poorly studied genes.

**Data Availability Statement:** All relevant data are within the paper and its Supporting Information files.

**Funding:** This research was funded by Mexico's National Council for Science and Technology, CONACYT, grant 284042. The funders had no role in study design, data collection and analysis, decision to publish, or preparation of the manuscript.

**Competing interests:** The authors have declared that no competing interests exist.

# Introduction

*Clostridioides difficile* is an opportunistic, gram-positive anaerobic bacillus responsible for the most common cause of diarrhea. *C. difficile* infection (CDI) is the most common healthcare-associated diarrhea and an important cause of morbidity and mortality [1]. In Mexico, an increase in the incidence of CDI was reported from 2012 to 2015, with a predominance of *C. difficile* ribotype (RT) 027 [2], which in addition to RT001, is the most common ribotype in this country [3]. When CDI recurs within the first 8 weeks of resolution of CDI symptoms, the episode is defined as recurrent CDI (R-CDI) [4, 5]. Several factors increase the risk of R-CDI, including successive episodes of disease (up to 65%) [1, 4], a previous history of CDI, and infection with the hypervirulent and epidemic strain *C. difficile* RT027 [6–9].

Some virulence factors have been associated with R-CDI, including sporulation and biofilm formation [10, 11]. Among sporulation evidence, mice infected with *C. difficile spo0A* knock-out mutant treated with oral vancomycin did not develop recurrence over 100 days of follow-up. In contrast, mice infected with *C. difficile* wild type strains developed R-CDI on the third day of follow-up, suggesting that spores may be a persistence factor [12].

Several studies have shown that *C. difficile* produces a biofilm that may protect the spores from antibiotics, including vancomycin [11, 13–16]. Moreover, increased expression of sporulation pathway genes (*sigH, spo0A*) was found in biofilm of *C. difficile* [17].

Biofilm formation has been associated with R-CDI, but the underlying transcriptomic patterns are largely unknown. Therefore, we analyzed the transcriptomic profile in an *in vitro* biofilm model of *C. difficile* RT001 and RT027 associated with recurrent CDI (R-CDI) and non-recurrent CDI (NR-CDI) to identify a gene expression signature that may favor the recurrence.

# Material and methods

## Ethics statement

This study was performed with approval from the Ethics Committee of the Hospital Universitario Dr. José Eleuterio González (approval number BI22-00003). The need for informed consent was waived because the study did not involve any physical intervention and was solely using the routinely collected clinical isolates. All data was analyzed anonymously.

## Strain collection, ribotyping and selection of clinical isolates

Clinical isolates were obtained from the Laboratory of Molecular Microbiology strain collection. Clinical isolates were recovered from patients with CDI or R-CDI, according to Surawicz *et al.* [4].

Bacterial DNA was extracted using the phenol-chloroform method [18]. As previously described, the intergenic region between 16S and 23S of rRNA genes was amplified by polymerase chain reaction (PCR) [19]. Strains *C. difficile* ATCC BAA-1805 (RT027) and *C. difficile* ATCC 9689 (RT001) were used as controls.

Twenty clinical isolates were selected: 10 from RT001 and 10 from RT027. From each ribotype, 5 isolates were from patients with R-CDI and 5 from patients with NR-CDI. Eighteen isolates were collected at the Hospital Civil of Guadalajara "Fray Antonio Alcalde" and two at the University Hospital Dr. José Eleuterio González).

## Biofilm production

Biofilm was prepared as previously reported [13], with some modifications. From a 7-day growth in Schaedler agar supplemented with 5% blood in anaerobic conditions (from each

isolate), a 0.5 McFarland suspension was adjusted and diluted 1: 100 in brain-heart infusion broth supplemented with 0.5% yeast extract and 0.1% L-cysteine (BHIS), 200 μl from each suspension were deposited on wells of a 96-well microplate (Sarstedt, Germany). Plates were incubated at 37˚C in anaerobiosis, and after 72 h, the medium was discarded. Plates were washed thrice with 200 μl of distilled water; the biofilm was fixed with 200 uL of methanol for 15 min and washed thrice, dried for 1 h at 60˚C and stained with 100 μl Hucker's crystal violet solution for 15 min at room temperature. Next, each well was washed four times with 200 μl distilled water and left to dry overnight at room temperature. Finally, 100 μl 99% ethanol (Sigma-Aldrich, USA) was added and left for 30 min. The optical density at 595 nm was measured in a multimode reader Cytation 1 (BioTek Instruments, Inc., USA). *C. difficile* ATCC BAA-1805 (RT027, strong biofilm producer) and *C. difficile* ATCC 9689 (RT001, strong biofilm producer) were used as controls. Assays for each clinical isolate were repeated eight times, and the BHIS broth was included as a blank. Biofilm production was classified according to criteria described by Vukovic D et al. [20].

## Biofilm and non-adherent cells for transcriptomic assays

Biofilm was prepared, as described in the previous section, but after incubation for 72 h, the medium was carefully discarded by sucking it through the cell wall. Eight replicates of each strain were pooled by scraping the bottom of the well with the tip of a micropipette, mixing with suction and ejecting with the micropipette, and depositing them in an Eppendorf tube. This mixture was analyzed as a biofilm.

For non-adherent cells obtention, biofilm was prepared as described, with incubation of 24 h instead of 72 h. After 24 h, 100 μl of the non-adherent cells in the supernatant of eight wells were carefully taken with a micropipette, grouped in an Eppendorf tube and analyzed as non-adherent cells.

## RNA extraction

The RNA was extracted from each sample (biofilm or non-adherent cells, 8 wells from the plate grouped). Each sample was centrifuged at 14,000 rpm for 5 min and incubated with 200 μL Tris-HCl (10 mM, pH 8) and 50 mg/L of lysozyme at 37˚C for 1 h. After incubation, proteinase K (10 mg/L) was added, and the tubes were incubated at 55˚C for 1 h. RNA was extracted using the Trizol Max Bacterial RNA Isolation kit (Thermo Fisher Scientific, USA) and resuspended in 50 μL RNase-free water.

## Pools for microarray analysis

Eight pools of RNA were prepared with the RNA from each of the 20 strains (biofilm or non-adherent cells, 40 RNAs in total). Each pool included 5 strains with the same characteristics and conditions to minimize the variation of one single strain.

Pools of RNA were prepared according to a format that included 3 conditions: biofilm or nonadherent cells, ribotype (001 or 027), and recurrence or not of CDI.

Each of the eight pools comprised RNA from five strains with the same characteristics as follows: pool 1: nonadherent, RT001, NR-CDI; pool 2: nonadherent, RT001, R-CDI; pool 3: nonadherent, RT027, NR-CDI; pool 4: nonadherent, RT027, R-CDI; pool 5: biofilm-RT001-NR, CDI, pool 6: biofilm, RT001, R-CDI; pool 7: biofilm, RT027, NR-CDI, and pool 8: biofilm, RT027, R-CDI.

The concentration of RNA from each pool was adjusted to 25 ng/μL with equal contribution of each sample.

## Microarray design and processing of the microarrays

Expression microarrays were designed for this study from the complete transcriptome of the strains *C. difficile* 630 (RT012) (Gen Bank: GCA_000009205), *C. difficile* ATCC 9689 (RT001) (Gen Bank: GCA_001077535), and *C. difficile* R20291 (RT027) (Gen Bank: GCA_000027105). The designed microarray was arranged in an 8 × 15K microarray format (Agilent Technologies, Inc., USA) in which 3,858 common genes in all strains 192 genes were unique for the strain *C. difficile* ATCC 9689; 249 genes, for *C. difficile* R20291, were included. The Agilent one-color, microarray-based exon analysis low input quick amp WT labeling protocol (Agilent Technologies, Inc., USA) was followed according to the manufacturer's instructions [21].

Each pool was separately labeled with Cy3 dye and hybridized on the designed microarray and shortly after being washed. Slides were scanned using the SureScan microarray scanner (Agilent Technologies, Inc., USA). Subsequently, data were extracted using the Agilent feature extraction software (Agilent Technologies, Inc., USA) [21].

## Microarrays bioinformatics analysis

The data were processed using the gene expression analysis platform v.0.4 software (also known as GEAP) [22]. Quantile type normalization was performed with the Linear Models for Microarrays Data (LIMMA) tool, and the quality analysis was conducted with arrayQualityMetrics.

Gene expression of all eight pools was compared with LIMMA using the Student's t-test with a statistical significance of ≤0.01 and a logarithmic fold change (LogFC) of 1.5. For multiple comparisons, the Benjamini-Hochberg correction was used.

Differentially expressed genes (DEGs) between pools were visualized with a heat map using GraphPad Prism v9.3.0. Software. Functional annotation was performed with the DAVID Knowledgebase Query Tool [23], and National Center for Biotechnology Information; protein interaction and enriched pathways were evaluated in the Gene Ontology (GO) and Encyclopedia of Genes Kyoto Genes and Genomes (KEGG) in String and ShinyGO v0.75 online tool [24].

## Validation of microarray results by qRT-PCR

Microarray results were validated by qRT-PCR on some selected genes using primers designed in this study (z 1). Two housekeeping genes (*rpsJ* and *rrs*) were used for normalization [25]. Retrotranscription was performed using 1 μL of SuperScript III 200U/μL (Invitrogen, USA), 4 ng random primers (Invitrogen, USA), 400 μM dNTP mix, and 125 ng total RNA from each of the pools for a final volume of 25 μL; the manufacturer recommended conditions were followed [26].

For qRT-PCR, 1 U Taq DNA polymerase (Invitrogen, USA), 400 μM dNTPs, 3.5 mM $MgCl_2$, 1× $NH_4$ reaction buffer, 3.125 μL 12.5× SYBR Green I (Sigma Aldrich, USA), 200 nM of each primer (except for *rrs*, 50 nM), and 160 ng cDNA were added for a final volume of 25 μL. qPCR conditions were as follows: 1 cycle at 95˚C for 2 min, then 35 cycles at 95˚C for 30 s, 60–62˚C (Table 1) for 30 s, and 72˚C for 25 s. Reactions were carried out using a CFX96 Touch Real-Time PCR Detection System thermal cycler (Bio-Rad, Korea). The change in the expression of each target gene was determined using the Delta-delta Ct Pfaffl relative quantification method [27], and the results were expressed as the mean relative expression ± standard deviation.

## Statistical analysis

The relative gene expression results from eight pools were compared using variance and post hoc Bonferroni tests. All analyses were performed using the statistical program IBM SPSS statistics v.25, with a p-value less than 0.01 for relative expression in qPCR.

**Table 1. Primers used for qPCR, expected product size and annealing temperature.**

| Gene | Primer (5´→ 3´) | Product (bp) | Annealing temperature (˚C) |
|------|-----------------|--------------|----------------------------|
| CAJ67607 | F-GTAGACCAAATAAGCCCAAAACCTAC | 136 | 62 |
| | R-CACCTAGTTTATCAAGACCTCCAAC | | |
| CAL68046 | F-GGGAAATTATTGGATGCTCGG | 210 | 62 |
| | R-CTCCTGCTATCTCATTACGCTCA | | |
| CAJ70248 | F-GCTGTTGCTGCTATATTTAATCCTG | 187 | 60 |
| | R-CAGCACTTTCACTTGTTGGTTC | | |
| CAJ67384 | F-ACAACGAAACCAGTAGGAGGT | 168 | 60 |
| | R-AGCAAGACAAGGAGTTCCCAT | | |
| CAJ68100 | F-GAACTACAACAACAAAGTCTTGAA | 238 | 62 |
| | R-CTGCTTGACTACTTTCGCCAT | | |
| *rpsJ* | F-GATCACAAGTTTCAGGACCTG | 151 | 60 |
| | R-GTCTTAGGTGTTGGATTAGC | | |
| *rrs* | F-GGGAGACTTGAGTGCAGGAG | 120 [25] | 60 |
| | R-GTGCCTCAGCGTCAGTTACA | | |

## Results

### Clinical isolates and analysis

All 027 strains were strong biofilm producers. Among RT 001 clinical isolates, five isolates were strong producers, 3 were moderate producers, and 2 clinical isolates were no biofilm producers. Biofilm production and ribotyping results are shown in Fig 1 and S1 Table.

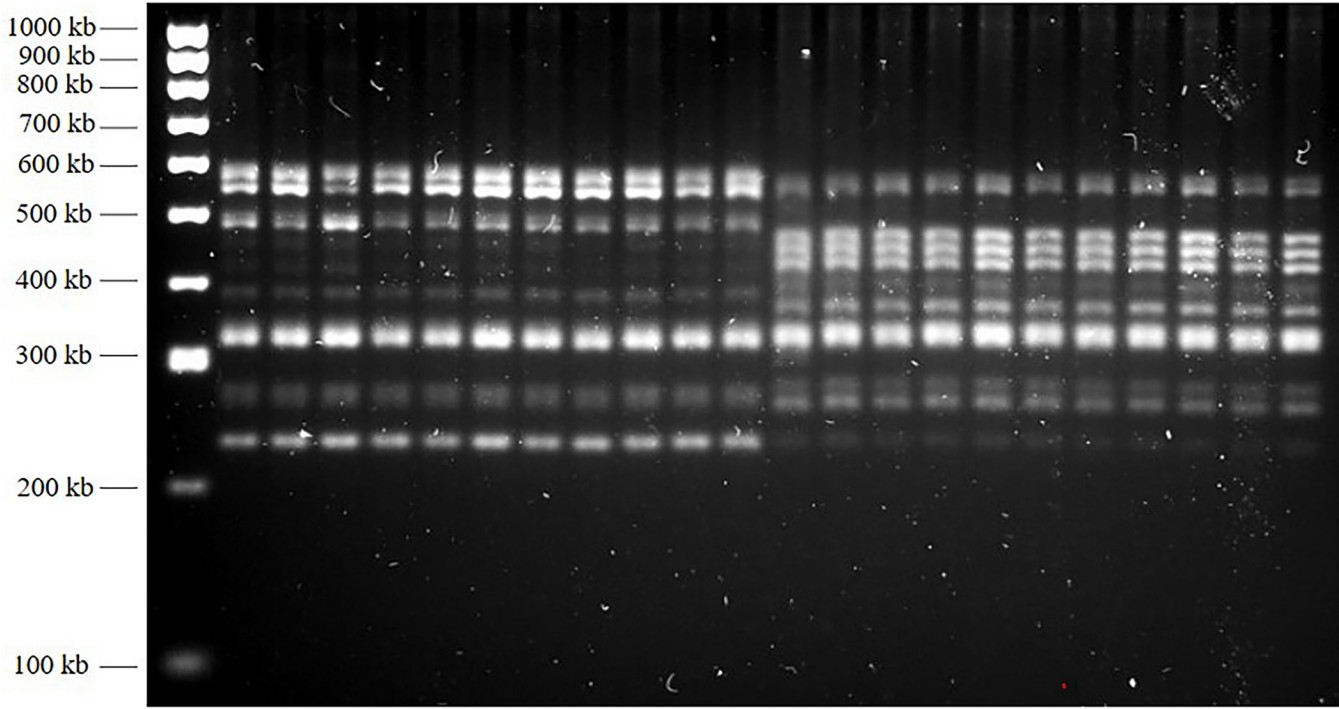

**Fig 1. Ribotyping of *Clostridioides difficile* strains.** Line 1: molecular weight marker. Line 2: Control strain ATCC 9689 (RT001). Line 13: ATCC BAA 1805 (RT027). Lines 3–12 and 14–23: clinical isolates.

## Microarrays quality and analysis

The eight pools yielded RNAc and specific activity greater than reference parameters (>0.825 vs. 1.8195 μg and 15 vs. 20.3365 pmol Cy3 per μg of RNAc), respectively. Gene expression of unique pools and combination of pools was compared.

## Pool 1 (nonadherent, RT001, NR-CDI) vs. pool 5 (biofilm, RT001, NR-CDI)

When we compared nonadherent cells vs. biofilm among RT001, NR-CDI strains, 149 genes were found to be overexpressed and 47 underexpressed from biofilm (Fig 2; S2 Table). The higher overexpression was detected for *CAJ70248* (LogFC 2.909), which encodes an exosporium collagen-like glycoprotein BclA3. Conversely, the most underexpressed was *CAJ67148* (LogFC, −1.955), which encodes an ABC-type transport system. Enriched pathways on gene ontology (GO) were associated with arginine metabolism, ethanolamine, organonitrogen compounds, small molecules, amines, the phosphotransferase system (PTS), and the transmembrane, carbohydrate, and organic substance transport pathways (Fig 3; S3 Table).

## Pool 2 (nonadherent, RT001, R-CDI) vs. pool 6 (biofilm, RT001, R-CDI)

The differential expression analysis between pool 2 and 6 revealed 56 overexpressed genes and 18 underexpressed from biofilm (Fig 2; S4 Table). The higher overexpression was detected in the *CAJ67274* gene (LogFC, 2.268), which encodes for a dependent enzyme of the alanine/ornithine racemase family, whereas the most underexpressed gene was *CAJ70208* (LogFC, −−3.188), which encodes an uncharacterized protein.

The most enriched pathways in GO were associated with the transport of carbohydrates, organic substances, and transmembrane carbohydrates. Additional enriched pathways were associated with metabolism among phosphoenolpyruvate-dependent sugar PTS and small molecule catabolic process (Fig 3; S3 Table).

## Pool 3 (nonadherent, RT027, NR-CDI) vs. pool 7 (biofilm, RT027, NR-CDI)

In this comparison, 45 genes were overexpressed, and 25 were underexpressed from biofilm (Fig 2; S5 Table). The most overexpressed gene was *CAJ67274* (LogFC, 2.268), which encodes a dependent enzyme of the alanine/ornithine racemase family, and the least underexpressed gene was *CAJ70208* (LogFC, −3.188), which encodes an uncharacterized protein. Two enriched pathways were detected, one associated with sporulation (genes, 4; FDR, 0.0010) and the other related to ascorbate and aldarate metabolism (KEGG; genes, 2; FDR, 0.0216) (Fig 3; S3 Table).

## Pool 4 (non-adherent, RT027, R-CDI) vs. pool 8 (biofilm, RT027, R-CDI)

When comparing DEGs in pools 4 and 8, 42 genes were overexpressed, and 22 were underexpressed in biofilm (Fig 2; S6 Table). The most overexpressed gene was *CAJ69029* (LogFC, 2.776), which encodes a sporulation membrane protein YtaF, and the more underexpressed gene was *CAJ67779* (LogFC, −2.52), which encodes a hypothetical protein. Moreover, a GO-enriched sporulation pathway (Gene 3; FDR 0.0411) was detected (Fig 3; S3 Table).

## Differentially expressed genes in biofilm in recurrent and non-recurrent strains (ribotype 001)

From the results obtained from the comparisons of Pool 1 (nonadherent, RT001, NR-CDI) vs. pool 5 (biofilm, RT001, NR-CDI) and Pool 2 (nonadherent, RT001, R-CDI) vs. pool 6 (biofilm,

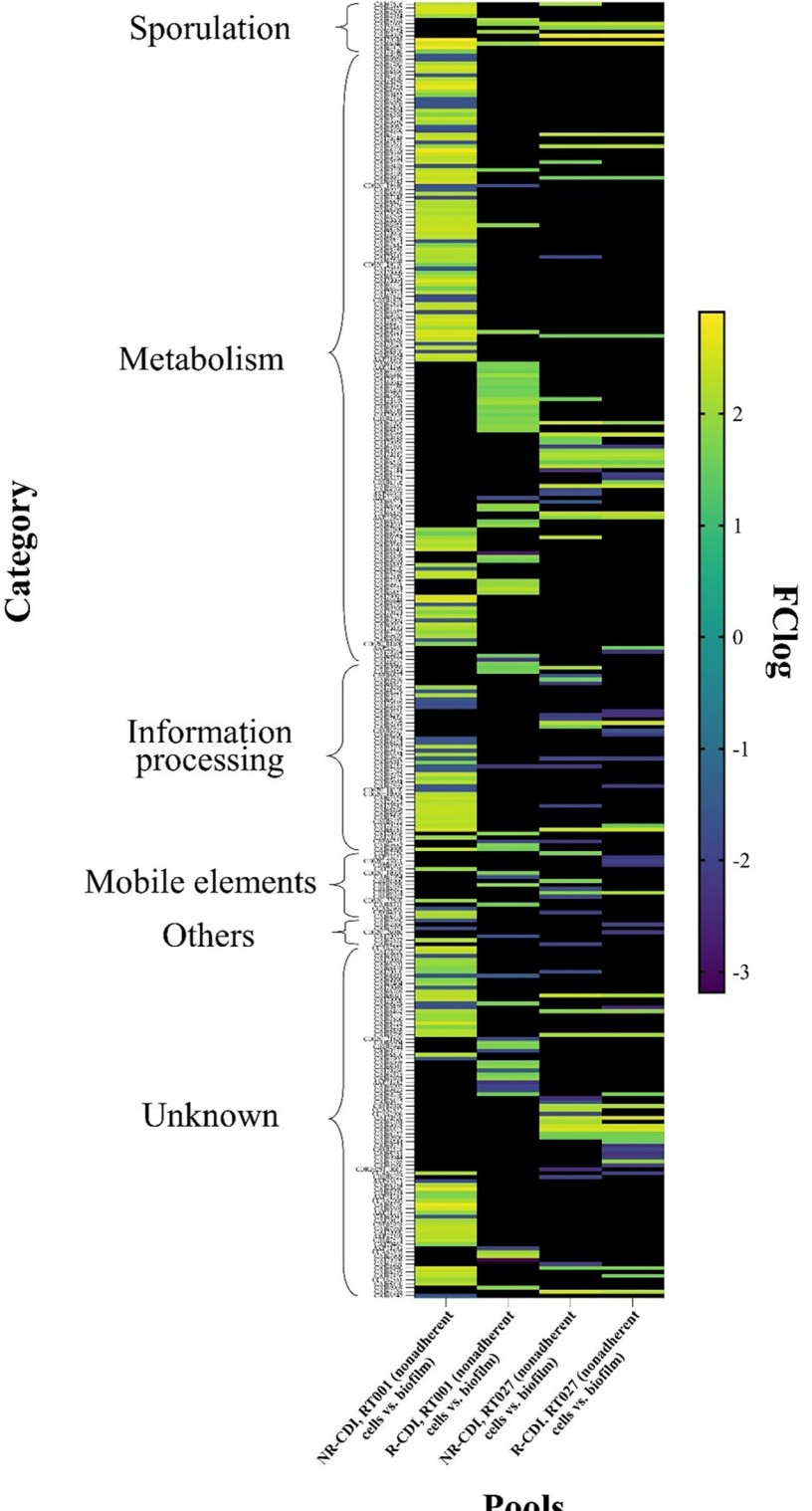

**Fig 2. Heat map of transcriptomic changes in biofilm at RT001 and 027 associated with NR-CDI and R-CDI.** RNA from nonadherent cells (RT001 and 027) associated with R-CDI and NR-CDI (eight pools) were labeled with Cy3 dye and hybridized on expression microarray. Slides were scanned and gene expression were compared with LIMMA using the Student's t-test. Only genes with FClog ≥ 1.5, ≤ -1.5 were considered; genes out of parameters are shown in black; overexpressed genes are in green and yellow colors; underexpressed genes are in blue.

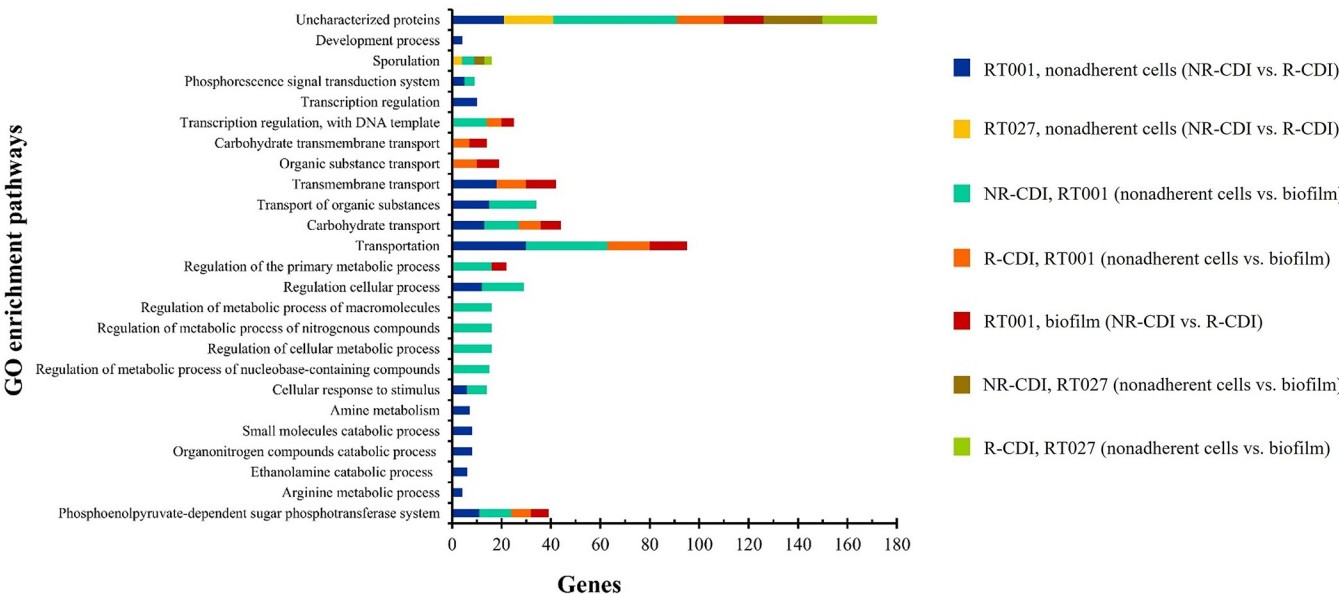

**Fig 3. Gene ontology (G.O.) enriched pathways of genes differentially expressed in nonadherent cells and biofilm of *C. difficile*.** Protein interaction and enriched pathways of DEGs found in nonadherent cells, and biofilm of RT001 and 027 or independent of ribotype, associated with R-CDI and NR-CDI were evaluated in the Gene Ontology (GO), Encyclopedia of Genes Kyoto Genes and Genomes (KEGG) in String and ShinyGO v0.75 online tool. FDR = <0.05.

RT001, R-CDI), heat maps were built, and 189 genes were unique for biofilm in non-recurrent strains, 8 genes were common, while 64 genes were unique for recurrent strains (50 genes overexpressed and 14 underexpressed) (Fig 2; S7–S9 Tables). The most overexpressed gene was *CAJ67269* (LogFC, 1.505), which encodes a 2-amino-4-ketopentanoate thiolase alpha sub-unit, and the least underexpressed gene was *CAJ70208* (LogFC, −3.188), which encodes an uncharacterized protein. Most of the enriched pathways in gene ontology were associated with transmembrane carbohydrates and general transport and other metabolites such as carbohydrates and organic substances, as well as a PTS metabolism pathway and pathways associated with the regulation of the cellular process and regulation of transcription (Fig 3; S3 Table).

Among 64 unique expressed genes in recurrent strains, 16 genes encode uncharacterized proteins, with two out of them having similarity to phage proteins phiC2 (*CBE03994*; query cover, 97%; identity, 96.24%) and phiCD211 (*CAJ69823*; query cover, 96%; identity, 75%) (S10 Table). No other similarities were detected.

## Differentially expressed genes in biofilm in recurrent and non-recurrent strains (ribotype 027)

From the results obtained from the comparisons of pool 3 (nonadherent, RT027, NR-CDI) vs. pool 7 (biofilm, RT027, NR-CDI) and pool 4 (nonadherent, RT027, R-CDI) vs. pool 8 (biofilm, RT027, R-CDI), heat maps were built and, where determined that 35 DEGs were unique for biofilm in non-recurrent strains, 35 genes were common, and 26 genes were unique for recur-rent strains (S11–S13 Tables).

For the latter, 7 were overexpressed, and 19 were underexpressed (Fig 2; S11 Table). The most overexpressed gene was *CAJ67765* (LogFC, 1.91), which encodes a sporulation mem-brane protein YtaF, and the least underexpressed gene was *CAJ69479* (LogFC, −2.52), which encodes an uncharacterized protein. No enriched pathways were found.

In addition, 11 genes encoded uncharacterized proteins. Amino acid sequences were searched using Blastp, and one of them had similarity to an amidohydrolase 3 (*CBE02518*; query cover, 78%; identity, 99.71%); two of them had similarity with a phage tail tube protein (*CBE04002*; query cover, 100%; identity, 100%) and a protein containing the C-terminal domain of phage (*CAJ67456*; query cover, 100%; identity, 98.79%) (S14 Table).

## Differentially expressed genes in biofilm in recurrent strains (ribotypes 001 and 027)

From the comparison between the 64 genes unique for recurrent strains of RT001 and 26 genes unique for recurrent strains of RT027, 63 genes were unique in RT001, 25 genes were unique in RT027, and 1 gene, *CAJ69276*, was common and overexpressed in both ribotypes (Fig 2). This gene encoded a probable membrane protein CAJ69276 (RT001: LogFC, 1.581; RT027: LogFC, 1.626). No protein homology was found in Blastp databases.

## Pools 1 (nonadherent, RT001, NR-CDI) and 3 (nonadherent, RT027, NR-CDI) vs. pools 5 (biofilm-RT001, NR-CDI) and 7 (biofilm, RT027, NR-CDI)

When comparing DEGs in both ribotypes of non-recurrent strains (nonadherent cells vs. biofilm), 13 genes were overexpressed and 4 underexpressed (Fig 4; S15 Table) in non-recurrent biofilm strains. After correction for multiple comparisons, no significant differences were observed.

## Pools 2 (nonadherent, RT001, R-CDI) and 4 (nonadherent, R027, R-CDI) vs. pools 6 (biofilm, RT001, R-CDI) and 8 (biofilm, RT027, R-CDI)

Comparing both ribotypes of recurrent strains (nonadherent cells vs. biofilm), we observed that 23 genes were overexpressed and 3 underexpressed (Fig 4; S16 Table) in recurrent biofilm strains. After Benjamini-Hochberg correction, genes were overexpressed: CAJ70148 (a putative dehydrogenase), *CAJ68100* (putative protein), *CAJ69725* (putative membrane protein), and one was underexpressed; *CAJ68151* (a segregation and condensation protein A) (adj. p = 0.05).

## Differentially expressed genes from both ribotypes in biofilm (non-recurrent and recurrent strains)

Finally, we compared DEGs in both ribotypes from biofilm (i.e., R-CDI vs. NR-CDI) and observed that 12 genes were unique in non-recurrent strains, 21 genes were unique in recurrent strains, and 4 genes were common (Fig 4; S17 and S18 Tables). For the latter, 18 were overexpressed, and 3 were underexpressed (Fig 4; S17 Table). After Benjamini-Hochberg correction, 3 genes were overexpressed (*CAJ70148*, a putative dehydrogenase: LogFC, 3.767; *CAJ68100*, putative protein: LogFC, 2.986; and *CAJ69725*, putative membrane protein: LogFC, 4.222) and 1 gene was underexpressed (*CAJ68151*, segregation and condensation protein A; LogFC, −2.877) (adj. p = $\leq$ 0.05) (S17 Table).

Among 21 unique expressed genes in recurrent strains, 11 genes encoding no characterized proteins were searched using Blastp, and 5 homologies were detected: similar to a type II secretion system (*CAJ68100*; query cover, 97%; identity, 46.11%), similar to a putative dehydrogenase (*CAJ70148*; query cover, 100%; identity, 100%) and a lactonase (*CAJ70148*; query cover, 100%; identity, 98.96%), similar to transposase (CBE06724; query cover, 100%; identity,

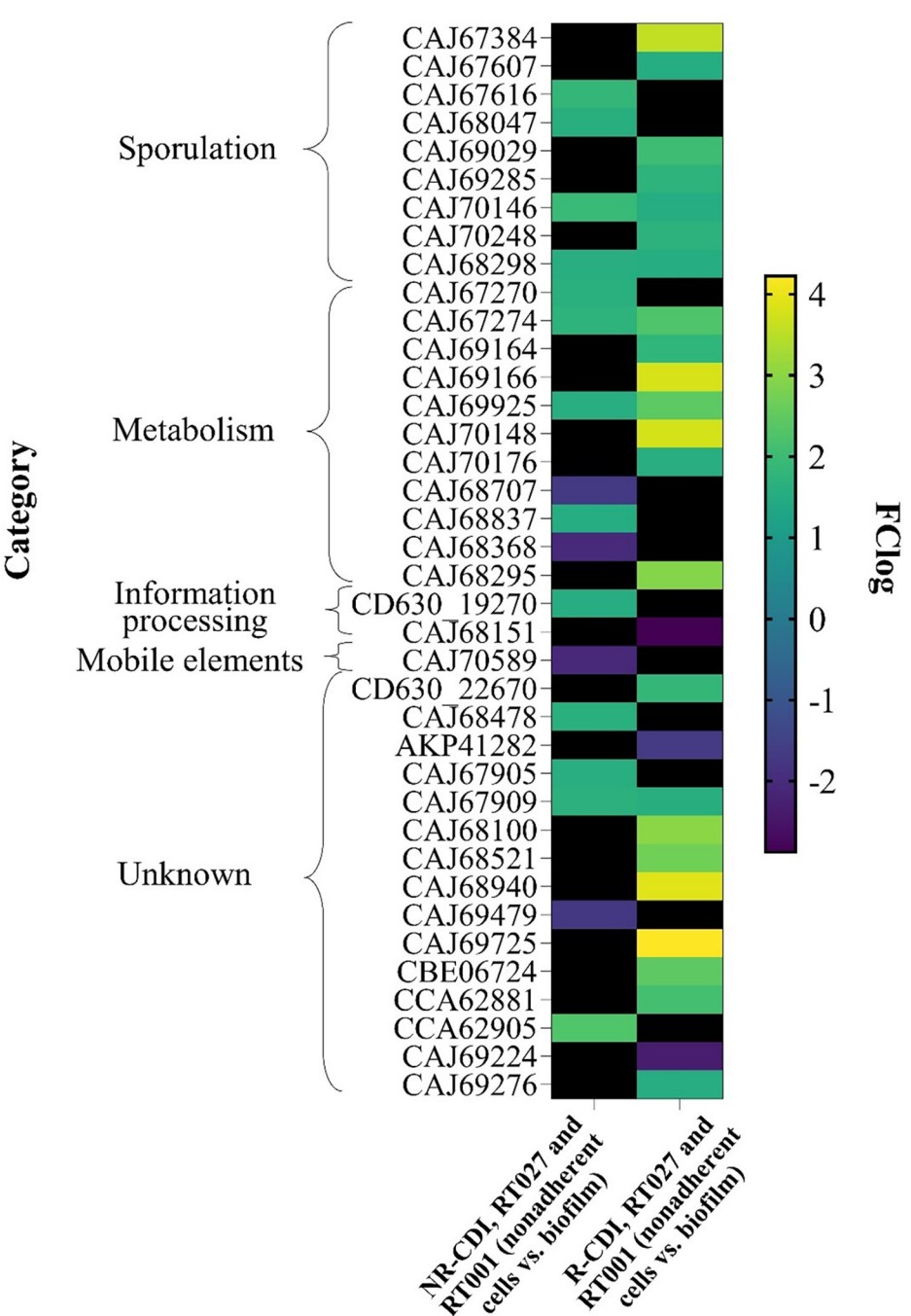

**Fig 4. Heat map of transcriptomic changes on biofilm associated with NR-CDI and R-CDI, independent of ribotype.** Data from DEGs on nonadherent cells and biofilm (R-CDI and NR-CDI strains independent of ribotype) was compared with LIMMA using the Student's t-test and Benjamini-Hochberg correction for multiple comparisons. Only genes with adj.p = <0.01 and FClog ≥ 1.5, ≤ -1.5 were considered; genes out of parameters are shown in black; overexpressed genes are in green and yellow colors; underexpressed genes are in blue colors.

75.44%), and a similar to a 5-bromo-4-chloroindolyl phosphate hydrolysis family protein (*CAJ69224*; query cover, 100%; identity, 95.10%).

## Validation of the results of the microarrays by RT-qPCR

Five overexpressed genes were selected to perform the validation of the results of the microarrays by means of RT-qPCR (LogFC, ≥2), including genes in the sporulation pathway and *CAJ70248* (associated with the recurrence of *C. difficile* R20291 [RT027]) (14). The overexpression of genes evaluated by qPCR is like what was found in microarrays. Therefore, the results are validated (S1 Fig).

## Discussion

Biofilm formation and changes in the genome of *C. difficile* have been associated with recurrence in CDI. However, the transcriptomic profile has been poorly studied, and the mechanisms underlying recurrence have not yet been completely understood. In this study, we analyzed the transcriptomic profile in an *in vitro* biofilm model of the *C. difficile* RT001 and RT027 ribotypes associated with R-CDI and NR-CDI. After multiple comparisons among groups involving features such as ribotype, biofilm formation, and recurrence of infection, we found several genes differentially expressed.

According to our results, four genes were differentially expressed (adj. p ≤ 0.05) in recurrence (pools 1 and 3 vs. pools 5 and 7) and biofilm (pools 5 and 7 vs. pools 6 and 8) evaluations. These genes correspond to a putative dehydrogenase (*CAJ70148*)—with high sequence homology to the YncE family protein and a lactonase—, a secretion type II system protein from the GspH (pseudopilins) family (*CAJ68100*), a putative membrane protein (*CAJ69725*), and the segregation and condensation protein A (*CAJ68151*); the first three were consistently overexpressed while the remaining underexpressed over comparisons. Taken together, differential expression of these genes appears to be essential to bacterial survival and communication functions.

The specific expression signature pointed to genes related to survival and biofilm maintenance/formation. However, the role of biofilm as a significant virulence and/or recurrence factor was rather contrasting among comparisons (i.e., in strains with biofilm formation but no recurrence). This is consistent with the notion that more than one mechanism (e.g., sporulation) is needed for the recurrence [28].

The putative dehydrogenase (*CAJ70148*) has a high sequence homology to the YncE family protein and a lactonase.

The *yncE* gene encodes the YncE protein identified in *Escherichia coli* [29]. YncE is an essential protein composed of a seven-bladed β-propeller [30, 31]. Analysis of subcellular fractions showed that YncE is mostly located in the periplasm [30]. Also, YncE has been identified as an antigen in patients whose bile ducts contain *Salmonella enterica* serovar Typhi, and it has been proposed that it might play a role in bacterial survival in the biliary tract [32].

*CAJ70148* is also a potential lactonase; these enzymes have been shown to inhibit bacterial *quorum sensing* [33, 34] by hydrolysis of the ester bond in the lactone ring of acylated homoserine lactones (AHLs) [33, 35].

Some bacterial pathogens produce AHLs as signaling molecules in response to population density [36, 37]. Thus, by degrading AHLs, lactonases may curb *quorum sensing* to reduce the capability of bacterial pathogens to trigger virulent expression, which may inhibit the production of virulence factors and biofilm formation. However, since Gram-positive bacteria use auto-inducer peptides other than lactones [38], thus, a potential lactonase overexpression here is intriguing.

*CAJ68100* encodes a type II secretion system (T2SS) protein from the GspH family. Proteins of this family participate in the assembling of the pseudopilus. This system is crucial for the survival of pathogen Gram-negative bacteria through the transport of toxins and several enzymes, including proteases, lipases and carbohydrate-active enzymes, to the cell surface or extracellular space of these bacteria [39]. Because secretion systems are virulence and pathogenicity factors [40], overexpression of *CAJ68100* in our study could be linked to both the recurrence and biofilm formation events in the ribotypes evaluated here.

*CAJ69725* is known to be a putative membrane protein. However, there is no additional information. As mentioned above, survival success often depends on extracellular communication, and this membrane protein's overexpression may be involved.

Noticeably, when both ribotypes of recurrent strains (biofilm vs. nonadherent cells) and both ribotypes from biofilm (R-CDI vs. NR-CDI) were compared, *CAJ68151* was underexpressed. *CAJ68151* (segregation and condensation protein A) has a role in chromosomal partition during cell division and was underexpressed in the biofilm of and in R-CDI strains. We speculate that this fact could be a result of a "stationary state" where persistent cells limit their metabolism (slow-growing) or forfeit propagation in the presence of lethal threats [41]. This landscape describes biofilm features and may explain recurrence in bacteria such as *Escherichia coli* [41]. Thus, it is tempting to visualize the down expression of this gene as a recurrence driver in *C. difficile*.

The bioinformatic analysis revealed that the most enriched pathways on biofilm recurrent strains were related to the transport of carbohydrates and organic substances, transmembrane carbohydrates transport, and the PTS (Fig 3; S3 Table). The PTS is the main bacterial system for carbohydrate assimilation [42] and has been associated with the absorption and metabolism of carbohydrates, carbon, nitrogen, and phosphorus, as well as the production of toxins and chemotaxis [43]. Members of this system were either overexpressed or underexpressed in our model, which may provide advantages to *C. difficile*. In fact, some bacteria, such as *Enterococcus faecalis*, can coordinate their metabolism to escape from the immune system [44], decreasing the metabolism of carbohydrates (mannose and fructose) by the PTS1 pathway and changing primary carbon resources for macrophages impairing the ability of these to clear intracellular *E. faecalis*; consequently, it decreases the polarization M1 and reduces nitric oxide production affecting immune response [45]. Furthermore, the downregulation of metabolism pathways (e.g., PTS, carbon catabolism, tricarboxylic acid pathway, and pentose phosphate pathway) [46] can cause an ATP depletion associated with persistent bacteria biofilm formation in *S. aureus* [47]. In addition, Kulecka et al. determined that groups of genes related to metabolism pathways, such as oxidative phosphorylation, are common in the genome of recurrence-related strains of *C. difficile* [48].

Despite the statistical limitations in our microarray study, as several genes were underestimated after the Benjamini-Hochberg correction, the expression signature pointed to genes related to maintaining biofilm, survival, and likely favor recurrence.

Therefore, it deserves further attention, and studies with more biological replicates are guaranteed.

Within DEGs without statistical significance in microarrays, *CAJ70248* was overexpressed in recurrent strains on nonadherent cells (RT001) (S3 Table). This protein encodes a protein that forms hair-like structures of the exosporium of *C. difficile* 20291 (RT027) and participates in adhesion [49] to fibronectin (α5β1 subunit), vitronectin [50], and integrin (αvβ1 subunit) to be internalized in colonocytes. This process may facilitate the immune response and antibiotic therapy evasion, thus favoring R-CDI [10].

Similarly, *CAJ67607* (coding the SpoVAC protein), *CAJ66946* (SpoIIID transcriptional regulator of sporulation protein), and *CAJ67428* (CotJA spore coat–associated protein) (Fig 2; S4

Table (on biofilm in RT001 recurrent strains), were also overexpressed. In the case of RT027 recurrent strains, genes such as *CAJ68046* (stage III sporulation protein AA) were overexpressed, while *CAJ67607* (SpoVAC) and *CAJ69029* (sporulation membrane protein YtaF) were underexpressed (Fig 2; S6 Table). In addition to sporulation genes, some virulence factors were detected to be differentially expressed. For example, *flgB*—which encodes a flagellar basal body rod protein—was underexpressed on biofilm in R-CDI strains (RT027) (Fig 2; S6 Table). Another virulence factor found in the present study is *CD630_26040*, which encodes a fragment of ADP-ribosyltransferase CdtAB [51–53]. This gene was underexpressed on biofilm in R-CDI strains (RT027) compared with nonadherent cells (Fig 2; S6 Table). The presence of genes encoding *C. difficile* toxin (CDT) can predict a recurrent episode in humans, and a combination of the detection of *tcdA*, *tcdB*, and CDT genes results in an odds ratio of 3.1 (95% confidence interval, 2.97–3.33) for R-CDI [54].

In our study, RT-qPCR experiments were performed to validate differential expressions in the microarrays. Genes were selected based on their expression and presence in recurrence.

Our results are the basis for further experiments, including some gene modifications- like gene deletion, gene knockdown or over-expression- to identify the role recurrence.

In *Bacillus* spp. and other clostridia, germination begins when a germinating molecule interacts with the germinative (Ger) receptor. *C. difficile* spores do not share homologs of the *gerA*, gerB, and *gerK* sprout receptors commonly recognized [55].

In *C. difficile*, the first step of germination is the binding of germinant to subtilin-like serine protease (CSP)-like proteins encoded at the *cspBAC* locus, where the *cspBA* and *cspC* genes that code for the CspBA and CspC are key regulators of germination signal transduction [56]. CspC functions as a receptor for bile salts. In addition, it transmits the signal through protein-protein interactions to CspB. Overexpression of *cspC* and *cspAB* has been found in a biofilm model of *C.difficile* 630Δerm growth at 37˚C in tryptone yeast extract supplemented with 0.1% sodium thioglycolate [57]. In our study, *the cspC gene was not overexpressed*, *possibly* due to the lack of thioglycolate in the medium used (BHIS).

The use of a pool of samples was included in this study to minimize the effects of individual patients and different strains (including the level of biofilm formation). We included the same patient characteristics, as this is the common denominator for the preparation of the pools.

It has been reported that in CDI-R, 70 to 81% is due to the same strain, and 15 to 30% is due to a different strain [58, 59]. Unfortunately, we have no strains collected in the initial CDI and the recurrent CDI to define whether the strain is the same. The use of pools is intended to minimize this effect.

Some ribotypes are frequently associated with recurrence, including RT012 [60], RT078, RT244 [61], and RT027 [7–9]. In contrast, other ribotypes have no clear association with recurrence, including RT001 RT106, RT017 and 018 [62–64]. We selected a ribotype clearly associated with recurrence and a second ribotype not clearly associated with recurrence. Our results provide an overview of the transcriptomic differences among strains associated with CDI-R and CDR-NR in a 3-day biofilm, depending on the ribotype.

The inclusion of other ribotypes will be of interest to clearly define the importance of ribotypes in the recurrence of infection.

In our study, *CAJ70148*, *CAJ68100*, *CAJ69725* and *CAJ68151* were differentially expressed in biofilm in strains associated with R-CDI. A significant limitation in the results is that some of these genes have not yet been studied, and the proteins they encode and the pathways in which they participate are unknown. Multiple investigations are required to determine the function of the unknown genes to elucidate the exact mechanisms.

In conclusion, genes related to survival, virulence, and sporulation were differentially expressed, which suggests an essential role in biofilm formation and recurrence. Moreover,

our model further supports the notion that other biological processes besides biofilm formation are involved in the recurrence. Finally, as long as we characterize unknown proteins, we will better understand the adaptations of *C. difficile* in R-CDI.

## Supporting information

**S1 Table. Classification of biofilm production in C. difficile.**
(DOCX)

**S2 Table. Differentially expressed genes in NR-CDI strains, RT001 (Pool 1, nonadherent, RT001, NR-CDI vs. Pool 5, biofilm, RT001, NR-CDI).**
(DOCX)

**S3 Table. Pathways enriched in gene ontology (GO) and Kyoto Encyclopedia of Genes and Genomes (KEGG).**
(DOCX)

**S4 Table. Differentially expressed genes in R-CDI strains, RT001 (Pool 2, nonadherent, RT001, R-CDI vs. Pool 6, biofilm, RT001, R-CDI).**
(DOCX)

**S5 Table. Differentially expressed genes in NR-CDI strains, RT027 (Pool 3, nonadherent, RT027, NR-CDI vs. Pool 7, biofilm, RT027, NR-CDI).**
(DOCX)

**S6 Table. Differentially expressed genes in R-CDI strains, RT027 Pool 4, non-adherent, RT027, R-CDI vs. Pool 8, biofilm, RT027, R-CDI).**
(DOCX)

**S7 Table.** Unique genes differentially expressed on biofilm R-CDI, RT001 strains. Pool 1 (nonadherent, RT001, NR-CDI) vs. Pool 5 (biofilm, RT001, NR-CDI) and Pool 2 (nonadherent, RT001, R-CDI) vs. Pool 6 (biofilm, RT001, R-CDI).
(DOCX)

**S8 Table. Unique genes differentially expressed on biofilm NR-CDI, RT001 strains.** Pool 1 (nonadherent, RT001, NR-CDI) vs. Pool 5 (biofilm, RT001, NR-CDI) and Pool 2 (nonadherent, RT001, R-CDI) vs. Pool 6 (biofilm, RT001, R-CDI).
(DOCX)

**S9 Table. Common genes differentially expressed on biofilm NR-CDI and R-CDI, RT001 strains.**
(DOCX)

**S10 Table. Identification of unique unidentified proteins by Blastp of C. difficile biofilm on biofilm R-CDI strains of RT001.** Pool 1 (nonadherent, RT001, NR-CDI) vs. Pool 5 (biofilm, RT001, NR-CDI) and Pool 2 (nonadherent, RT001, R-CDI) vs. Pool 6 (biofilm, RT001, R-CDI).
(DOCX)

**S11 Table. Unique genes differentially expressed in biofilm of R-CDI RT027 strains.** Pool 3 (nonadherent, RT027, NR-CDI) vs. Pool 7 (biofilm, RT027, NR-CDI) and Pool 4 (nonadherent, RT027, R-CDI) vs. Pool 8 (biofilm, RT027, R-CDI).
(DOCX)

**S12 Table. Unique genes differentially expressed on biofilm NR-CDI, RT027 strains.** Pool 3 (nonadherent, RT027, NR-CDI) vs. Pool 7 (biofilm, RT027, NR-CDI) and Pool 4 (nonadherent, RT027, R-CDI) vs. Pool 8 (biofilm, RT027, R-CDI).
(DOCX)

**S13 Table. Common genes differentially expressed on biofilm NR-CDI and R-CDI, RT027 strains.** Pool 3 (nonadherent, RT027, NR-CDI) vs. Pool 7 (biofilm, RT027, NR-CDI) and Pool 4 (nonadherent, RT027, R-CDI) vs. Pool 8 (biofilm, RT027, R-CDI).
(DOCX)

**S14 Table. Identification of unique unidentified proteins by Blastp in C. difficile biofilm of R-CDI, RT027 strains.** Pool 3 (nonadherent, RT027, NR-CDI) vs. Pool 7 (biofilm, RT027, NR-CDI) and Pool 4 (nonadherent, RT027, R-CDI) vs. Pool 8 (biofilm, RT027, R-CDI).
(DOCX)

**S15 Table. Differentially expressed genes in RT001 and RT027, NR-CDI strains (nonadherent cells vs. biofilm).** Pool 1 (nonadherent, RT001, NR-CDI) and Pool 3 (nonadherent, RT027, NR-CDI) vs. Pool 5 (biofilm, RT001, NR-CDI) and Pool 7 (Biofilm, RT027, NR-CDI).
(DOCX)

**S16 Table. Differentially expressed genes in RT001 and RT027, R-CDI strains (nonadherent vs. biofilm).** Pool 2 (nonadherent, RT001, R-CDI) and Pool 4 (nonadherent, R027, R-CDI) vs. Pool 6 (biofilm, RT001, R-CDI) and Pool 8 (biofilm, RT027, R-CDI).
(DOCX)

**S17 Table. Unique differentially expressed genes biofilm R-CDI, independent of ribotype.**
(DOCX)

**S18 Table. Unique differentially expressed genes biofilm NR-CDI, independent of ribotype.**
(DOCX)

**S1 Fig. Graph of relative expression of the 8 pools of interest.** Bonferroni post hoc p≤ 0.01.
(JPG)

**S1 Raw image. Ribotyping of Clostridioides difficile strains.** Line 1: molecular weight marker. Line 2: Control strain ATCC 9689 (RT001). Line 13: ATCC BAA 1805 (RT027). Lines 3–12 and 14–23: clinical isolates. Fig 1 was created from this image. Image was captured using a BioRad Molecular Imager ChemiDoc XRS instrument.
(PDF)

## Acknowledgments

We thank MSc Ana Zarazua for her help with the microarray analysis and Lucy Acevedo for her technical assistance.

## Author Contributions

**Conceptualization:** Rayo Morfín-Otero, Elvira Garza-González.

**Data curation:** Daira Rubio-Mendoza, Carlos Córdova-Fletes.

**Formal analysis:** Daira Rubio-Mendoza, Carlos Córdova-Fletes.

**Funding acquisition:** Elvira Garza-González.

**Investigation:** Rayo Morfín-Otero.

**Methodology:** Daira Rubio-Mendoza, Carlos Córdova-Fletes, Adrián Martínez-Meléndez, Rayo Morfín-Otero, Héctor Jesús Maldonado-Garza, Elvira Garza-González.

**Project administration:** Elvira Garza-González.

**Resources:** Carlos Córdova-Fletes.

**Supervision:** Daira Rubio-Mendoza, Carlos Córdova-Fletes, Elvira Garza-González.

**Validation:** Daira Rubio-Mendoza, Carlos Córdova-Fletes, Elvira Garza-González.

**Writing – original draft:** Daira Rubio-Mendoza.

**Writing – review & editing:** Daira Rubio-Mendoza, Carlos Córdova-Fletes, Adrián Martínez-Meléndez, Rayo Morfín-Otero, Héctor Jesús Maldonado-Garza, Elvira Garza-González.

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
