## [Decision Letter · Decision Letter 0]

9 May 2023

PONE-D-23-03435Transcriptomic analysis of biofilm formation in strains of Clostridioides difficile associated with recurrent and nonrecurrent infection reveals potential candidate markers for recurrence.PLOS ONE

Dear Dr. Garza-González,

Thank you for submitting your manuscript to PLOS ONE. After careful consideration, we feel that it has merit but does not fully meet PLOS ONE’s publication criteria as it currently stands. Therefore, we invite you to submit a revised version of the manuscript that addresses the points raised during the review process.

We look forward to receiving your revised manuscript.

Kind regards,

Rajesh P. Shastry

Academic Editor

PLOS ONE

Journal Requirements:

2. Please make sure that all information entered in the 'Ethics Statement' section regarding ethics approval is also included in the Methods section of the manuscript.

 [copy in funding statement].     

6.  We suggest you thoroughly copyedit your manuscript for language usage, spelling, and grammar. If you do not know anyone who can help you do this, you may wish to consider employing a professional scientific editing service.

 Whilst you may use any professional scientific editing service of your choice, PLOS has partnered with both American Journal Experts (AJE) and Editage to provide discounted services to PLOS authors. Both organizations have experience helping authors meet PLOS guidelines and can provide language editing, translation, manuscript formatting, and figure formatting to ensure your manuscript meets our submission guidelines. To take advantage of our partnership with AJE, visit the AJE website (http://aje.com/go/plos) for a 15% discount off AJE services. To take advantage of our partnership with Editage, visit the Editage website (www.editage.com) and enter referral code PLOSEDIT for a 15% discount off Editage services. If the PLOS editorial team finds any language issues in text that either AJE or Editage has edited, the service provider will re-edit the text for free.

7. PLOS ONE now requires that authors provide the original uncropped and unadjusted images underlying all blot or gel results reported in a submission’s figures or Supporting Information files. This policy and the journal’s other requirements for blot/gel reporting and figure preparation are described in detail at https://journals.plos.org/plosone/s/figures#loc-blot-and-gel-reporting-requirements and https://journals.plos.org/plosone/s/figures#loc-preparing-figures-from-image-files. When you submit your revised manuscript, please ensure that your figures adhere fully to these guidelines and provide the original underlying images for all blot or gel data reported in your submission. See the following link for instructions on providing the original image data: https://journals.plos.org/plosone/s/figures#loc-original-images-for-blots-and-gels.  

Reviewers' comments:

Reviewer's Responses to Questions

**Comments to the Author**

1. Is the manuscript technically sound, and do the data support the conclusions?

Reviewer #1: Partly

Reviewer #2: Partly

2. Has the statistical analysis been performed appropriately and rigorously? 

Reviewer #1: Yes

Reviewer #2: I Don't Know

3. Have the authors made all data underlying the findings in their manuscript fully available?

Reviewer #1: Yes

Reviewer #2: Yes

4. Is the manuscript presented in an intelligible fashion and written in standard English?

Reviewer #1: Yes

Reviewer #2: Yes

5. Review Comments to the Author

Reviewer #1: 1. This is good attempt to investigate the role of differentially regulated genes during biofilm formation.

2. Objective, materials and methods and results have been described very clearly. Data analysis has been done in details and representation of data is concise. But the paper lacks any conclusive evidence about the role of genes in biofilm formation or R-CDI. There is more of speculation or information-oriented paper rather than pinning down the mechanisms for R-CDI.

3. RT-qPCR validation of few genes might give a hint towards the mechanism involved in biofilm formation, but orthogonal experiments with some gene modifications- like gene deletion, gene knockdown or over expression etc. are essential.

4. It is surprising that other proteins involved like bile acid receptor (CspC), germination receptor (like GerK etc.), which have been previously shown are not affected in the transcriptomic analysis.

Reviewer #2: Dear authors,

Many thanks for this interesting article which highlights the impact of biofilm formation on recurrence of C. difficile.

However, I would have some points that would be nice if they could be addressed.

I am not sure if a pool of probes is really a good way to evaluate this research question. Recurrence of C. difficile is complex but one common denominator are patients' characteristics as well. Because a strain did not cause clinically a recurrent disease does not automatically imply that biofilm formation is lower. By the way in ~50% of rCDI the patient has been infected with another strain he acquired from the environment. In the other 50% the strain persists.

The genetic diversity is rather low, since only RT001 and RT027 were included. I would suggest the testing on a broader variety of strains (including the CD630 global RT012 reference strain) and each strain for itself. If I have seen it correctly, there were no strains with low bioflim formation included?

Minor:

Please harmonize the Material and methods part: sometimes the manufacturer is given with city, then with the state etc.

Abstract: Please shorten the Material and method part, e.g. you may not give manufacturers name in this section.

"nonrecurrent" or non-recurrent? (also in the title), please check.

Some small English language editing can be considered.

I would insert a "space" before the references.

With the best regards.

6. PLOS authors have the option to publish the peer review history of their article (what does this mean?). If published, this will include your full peer review and any attached files.

Reviewer #1: No

Reviewer #2: No

---

## [Author Response · Author response to Decision Letter 0]

4 Jul 2023

Rajesh P. Shastry

Academic Editor

PLOS ONE

Dear Editor:

Please find below the response for comments kindly provided by reviewers to our Manuscript PONE-D-23-03435 Transcriptomic analysis of biofilm formation in strains of Clostridioides difficile associated with recurrent and non-recurrent infection reveals potential candidate markers for recurrence. The professional service Scribbr edited our manuscript (https://order.scribbr.es/, order 2417109, revised by Debra, https://www.scribbr.com/about-us/editors/?_ga=2.73435346.1109912598.1685462982-2142197220.1685462982)

This research was funded by Mexico’s National Council for Science and Technology, CONACYT, grant 284042. The funders had no role in study design, data collection and analysis, decision to publish, or preparation of the manuscript.

Regarding figures, we are sending the original un crop image.

We hope that this revision will answer the concerns and suggestions. 

Yours sincerely,

The Authors

Reviewer #1: 1. 

2. Objective, materials and methods and results have been described very clearly. Data analysis has been done in details and representation of data is concise. But the paper lacks any conclusive evidence about the role of genes in biofilm formation or R-CDI. There is more of speculation or information-oriented paper rather than pinning down the mechanisms for R-CDI.

Response: In our study, we detected that CAJ70148, CAJ68100, CAJ69725 and CAJ68151 genes were differentially expressed in biofilm and strains associated with R-CDI. A significant limitation in the results is that some of these genes have not yet been studied, and the proteins they encode and the pathways in which they participate are unknown. To elucidate the exact mechanisms, multiple investigations are required to determine the function of the unknown genes. Our results are the basis for further experiments, including some gene modifications- like gene deletion, gene knockdown or over-expression- to identify the role recurrence. This comment was added in the new manuscript, Discussion section. These comments were added in the new manuscript. 

3. RT-qPCR validation of a few genes might give a hint towards the mechanism involved in biofilm formation, but orthogonal experiments with some gene modifications- like gene deletion, gene knockdown or over-expression etc. are essential.

Response: In our study, RT-qPCR experiments were performed to validate differential expressions in the microarrays. Genes were selected based on their expression and presence in recurrence. 

Our results are the basis for further experiments, including some gene modifications- like gene deletion, gene knockdown or over-expression- to identify the role recurrence. This comment was added in the new manuscript, Discussion section. 

4. It is surprising that other proteins involved like bile acid receptor (CspC), germination receptor (like GerK etc.), which have been previously shown are not affected in the transcriptomic analysis.

Response: Germination begins when a germinating molecule interacts with the germinative (Ger) receptor. C. difficile spores do not share homologs of the gerA, gerB, and gerK spreout receptors commonly recognized in Bacillus spp. and other clostridia (Clin Microbiol Infect. 2018;24(5):476–82).

In C. difficile, the first step of germination is the binding of germinants to subtilin-like serine protease (Csp)-like proteins encoded at the cspBAC locus, where the cspBA and cspC genes that code for the CspBA and CspC which are key regulators of germination signal transduction (Trends Microbiol. 2020;28(9):744–52). 

CspC functions as a receptor for bile salts. In addition to activating and regulating, it transmits the signal through protein-protein interactions to CspB. Overexpression of cspC and cspAB has been found in a biofilm model of C.difficile 630Δerm grown at 37°C in continuous-flow glass micro-fermentor in Tryptone Yeast Extract supplemented with 0.1% sodium thioglycolate incubating for 72 h (Front Microbiol. 2018 Sep 12;9:2084). The cspC gene was not overexpressed in the present work, probably because our methodology, the medium used (BHIS medium), had no thioglycolate. These comments were added to the new manuscript, Discussion section. 

Reviewer #2: 

I am not sure if a pool of probes is really a good way to evaluate this research question. Recurrence of C. difficile is complex but one common denominator are patients' characteristics as well. Because a strain did not cause clinically a recurrent disease does not automatically imply that biofilm formation is lower. 

Response: The use of pool of samples was included to minimize the effects of individual patients, and different strains. We included the same patients characteristics, as this is the common denominator for preparation of the pools. We agree with the reviewer that the association of a strain with recurrent disease does not imply that biofilm formation is lower. Indeed, we observed variation in the level of biofilm production among strains associated with recurrent infection. The use of pools is intended to minimize this effect. This comment was added to the new manuscript.

By the way in ~50% of rCDI the patient has been infected with another strain acquired from the environment. In the other 50% the strain persists.

Response: It has been reported that in CDI-R, 70 to 81% is due to the same strain and 15 to 30% is due to a different strain (J Infect Dis. 2014;209(9):1446–51., J Hosp Infect. 2015;90(2):108–16). Unfortunately, we have no strains collected in the initial CDI and the recurrent-CDI to define is the strain is the same or not. The use of pools is intended to minimize this effect. This comment was added to the new manuscript.

The genetic diversity is rather low, since only RT001 and RT027 were included. I would suggest the testing on a broader variety of strains (including the CD630 global RT012 reference strain) and each strain for itself. 

Response: Some ribotypes are frequently associated with recurrence, including RT012 (Eur J Clin Microbiol Infect Dis. 2017;36(11):2251–8.), RT078, RT244 (S D Med. 2017;70(9):422–3), and RT027 (PLoS One. 2014;9(6)., J Clin Microbiol. 2012;50(12):4078–82., mSphere. 2018;3(3):1–8.). 

In contrast, other ribotypes have no clear association with recurrence, including RT001, RT106, RT017 and 018 (APMIS. 2013;121(2):153-7.,J Hosp Infect. 2017;95(4):394-399. PLoS One. 2016;11(11):e01661595–17). We selected a common ribotype clearly associated with recurrence and a second ribotype not clearly associated whit. Our results provide an overview of the transcriptomic differences among strains associated with CDI-R and CDR-NR in a 3-day biofilm, depending on the ribotype. 

The inclusion of other ribotypes will be of interest to clearly define the importance of ribotypes in the recurrence of infection. This comment was added in the new manuscript, Discussion section.

If I have seen it correctly, there were no strains with low bioflim formation included?

Response: Biofilm production was quantified after 3 days of incubation. Of all the strains included in the study, all the strains included RT027 associated with CDI-NR and CDI-R were strong biofilm producers. In the case of RT001, of the strains associated with CDI-NR, 1 was a strong producer, 2 were moderate producers, and 2 were non-producers, and for the strains associated with CDI-R, 5 were strong producers. (S1 Table. Classification of biofilm production in C. difficile).

Minor:

Please harmonize the Material and methods part: sometimes the manufacturer is given with city, then with the state etc.

Response: Manufacturers were harmonized in the Material and methods section. The country is included in manufacturer details.

Abstract: Please shorten the Material and method part, e.g. you may not give manufacturers name in this section.

Response: Abstract was shortened in the method section

"nonrecurrent" or non-recurrent? (also in the title), please check.

Response: The term non-recurrent is now used I all the manuscript

Some small English language editing can be considered.

Response: English language was performed

I would insert a "space" before the references.

Response: A space was added before the references

---

## [Decision Letter · Decision Letter 1]

24 Jul 2023

Transcriptomic analysis of biofilm formation in strains of Clostridioides difficile associated with recurrent and nonrecurrent infection reveals potential candidate markers for recurrence.

PONE-D-23-03435R1

Dear Dr. Garza-González,

We’re pleased to inform you that your manuscript has been judged scientifically suitable for publication and will be formally accepted for publication once it meets all outstanding technical requirements.

Kind regards,

Rajesh P. Shastry

Academic Editor

PLOS ONE

Additional Editor Comments (optional):

I have included the some minor changes as suggested by the reviewer. Please address the changes in the manuscript before finalization.

Reviewers' comments:

Reviewer's Responses to Questions

**Comments to the Author**

1. If the authors have adequately addressed your comments raised in a previous round of review and you feel that this manuscript is now acceptable for publication, you may indicate that here to bypass the “Comments to the Author” section, enter your conflict of interest statement in the “Confidential to Editor” section, and submit your "Accept" recommendation.

Reviewer #1: All comments have been addressed

2. Is the manuscript technically sound, and do the data support the conclusions?

Reviewer #1: Yes

3. Has the statistical analysis been performed appropriately and rigorously? 

Reviewer #1: Yes

4. Have the authors made all data underlying the findings in their manuscript fully available?

Reviewer #1: Yes

5. Is the manuscript presented in an intelligible fashion and written in standard English?

Reviewer #1: Yes

6. Review Comments to the Author

Reviewer #1: 1. All the comments during the first round were carefully addressed and the manuscript is ready for acceptance.

2. For further studies, it would be interesting to see how the phenotypes of R-CDI and NR-CDI strains looks like in different media. BHIS is a very rich medium and C. difficile might not always encounter rich environmental conditions in the gut.

Suggestions for few minor changes are below:

a) Line 63: Change ‘nonmutant’ to wild type (WT).

b) Line 99: “After, 200 uL from each suspension……”. Delete the word “After”.

c) Line 102: “… the biofilm was fixed with 200 uL with methanol…..” Edit the sentence as “…. Was fixed with 200 uL of methanol…”

7. PLOS authors have the option to publish the peer review history of their article (what does this mean?). If published, this will include your full peer review and any attached files.

Reviewer #1: No

---

## [Editor Report · Acceptance letter]

27 Jul 2023

PONE-D-23-03435R1 

Transcriptomic analysis of biofilm formation in strains of *Clostridioides difficile* associated with recurrent and non-recurrent infection reveals potential candidate markers for recurrence. 

Dear Dr. Garza-González:

I'm pleased to inform you that your manuscript has been deemed suitable for publication in PLOS ONE. Congratulations! Your manuscript is now with our production department. 

Kind regards, 

on behalf of

Dr. Rajesh P. Shastry 

Academic Editor

PLOS ONE